# “That You Just Know You’re Not Alone and Other People Have Gone through It Too.” Eating Disorder Recovery Accounts on Instagram as a Chance for Self-Help? A Qualitative Interview Study among People Affected and Self-Help Experts

**DOI:** 10.3390/ijerph191811334

**Published:** 2022-09-09

**Authors:** Vanessa Wenig, Hanna Janetzke

**Affiliations:** 1Institute of Health and Nursing Science, Charité–Universitätsmedizin Berlin, Corporate Member of Freie Universität Berlin and Humboldt-Universität zu Berlin, Augustenburger Platz 1, 13353 Berlin, Germany; 2Department of Health, Nursing, Management, University of Applied Sciences Neubrandenburg, 17033 Neubrandenburg, Germany

**Keywords:** eating disorder recovery, self-help, recovery stories, social media, Instagram, qualitative interview, thematic analysis, qualitative content analysis

## Abstract

In addition to the professional treatment of eating disorders, the use of self-help groups has become increasingly important. Social media offers new possibilities for self-help, not only as online groups but also in increased access to recovery stories of people with similar diseases. People with eating disorders use the internet and social media depending on their motivation in different ways. Eating disorder recovery stories on social media have not yet been systematically used in treatment as appropriate guidelines are still lacking. This study provides an initial insight into the possibilities of using social media for self-help for eating disorders. Due to the exploratory nature, a qualitative design was used, combining interviews with people who have a recovery account on Instagram (*n* = 6) and self-help experts (*n* = 2). The results show that recovery stories on Instagram could serve as door openers for further treatment, motivation for therapy, a first step towards behaviour change, and support for existing therapies. If affected people can cope with the self-protection strategies, they can use Instagram positively for themselves and their disease. Nevertheless, there is a risk of negative influence as well as a risk of content and time overload. Therapeutic personnel can use these results to improve existing support services.

## 1. Introduction

In addition to the professional treatment of mental diseases, the use of self-help groups has become increasingly important [1]. In Germany, self-help is defined as “… voluntary groupings of affected people… whose activities are directed towards the joint management of diseases, consequences of diseases and/or also psychological problems from which they are affected either themselves or as relatives. They are not run by professional… staff” [2]. Self-help is low-threshold, with good access, and helps to close existing gaps in care by shortening waiting times for a place in therapy or a stay in hospital [3]. Self-help can also ensure the success of treatment for mental diseases and can increase the willingness to undergo therapy [3]. In the United Kingdom, self-help is used to reduce the number and/or length of therapeutic sessions [4]. In the international literature, the term “guided self-help” (GSH) has been established as a new form of self-help and therapy for eating disorders (ED) [5,6,7]. Such self-help programs vary widely in design. An exchange between patients, as in traditional self-help groups, usually does not take place. However, these GSH programs predominantly involve direct support from a healthcare professional [7] and are designed in a way that patients can follow specific manuals (in the form of a book, app, or internet-based program) independently [8]. In a German study, an ED self-help manual significantly shortened a following hospital stay and was thus more favourable compared to a waiting list [6]. GSH programs can also achieve long-term positive effects on pathology [7,9].

Moreover, Beveridge et al. [10] have highlighted the effectiveness of different treatment support in an Australian study: a peer mentoring program (PMP). The PMP focuses on the sharing of recovery experiences between a mentor (who used to be affected by an ED) and a mentee (who is currently affected by an ED). The PMP can reduce ED symptoms and improve the quality of life.

Internationally, internet-based programs have been increasingly used in the treatment of ED [11]. Internet-based programs can have the potential for ED treatment support or aftercare, and ED symptoms can be reduced [11], and internet-based therapy is superior to a waiting list [12]. Many of the online programs are based on cognitive behavioral therapy and are supervised by a psychological specialist. In some cases, internet-based programs can increase the motivation to change [13,14], and achieve long-term effects [13,15], or even ensure treatment success [16].

The trend towards internet-based ED support programs is clearly visible. In contrast, in Germany, only partially guided self-help for treatment support is recommended [17]. Evidence-based recommendations for non-guided self-help groups or other non-professional treatment support do not yet exist in Germany. However, younger patients are less able to identify with the traditional forms of self-help (circle of chairs) [18]. In addition, two German media studies [19,20] show that younger age groups (14–29 years) spend more time on social media like Instagram. The prevalence of ED symptoms is also high in this young age group. In Germany, the prevalence of ED symptoms in girls aged 11–17 years is 27.9% and in boys it is 12.1% [21]. Thus, self-help still must address new media to a greater extent. Moreover, the mortality rate of anorexia is particularly alarming—it is significantly higher than that of other mental diseases such as schizophrenia or depression [22]. As mentioned by Favotto et al. [23], computer-mediated communication has a two-sided impact on social, mental, emotional, and physical health. At the same time, it is also confirmed that the media can contribute to the development and adherence of ED and be responsible for a distorted body image [24,25,26,27,28,29]. In fact, simply looking at pictures of thin women has a significant negative impact on one’s own body image [30], and the use of social media platforms, such as Instagram and Snapchat, is significantly associated with increased ED behaviour in adolescent girls and boys [31].

However, there are further positive effects that are worth considering. For example, disclosing mental health issues on social media can have far-reaching mental health benefits [32]. Moreover, work by Walther and Boyd [33] on computer-mediated communication and social support shows that electronic support has become very popular for stigmatised individuals. The most important reasons are anonymity, interaction management, and immediate access [33]. In the world of social media, a virtual exchange between patients in the field of ED has developed in recent years. In addition to the disease-promoting pro-ana or pro-mia forums, in which patients glorify and jointly support their eating disorders [34], a positive online communication has developed in parallel: so-called recovery content. One can find ED recovery stories on the internet, in forums, in blogs, or on social media [35]. Such first-person accounts may promote patient empowerment for adaptation and healing [36], as well as hope, inspiration, and a sense of not being alone [37]. As noted by Smahelova et al. [38], people with ED use the internet and social media in very different ways, and their online use depends on their motivation. People with ED can use each platform exactly according to their motivation—either to become even sicker or for support during recovery [38]. Therapy motivation is an important aspect of successful treatment of ED. Several studies have shown [39,40,41] that a high level of motivation has a positive effect on the willingness to undergo treatment, the duration of therapy, and the reduction of pathological eating habits. Indeed, nonprofessional pro-recovery platforms are used by patients to look for realistic examples of the recovery process or to see what has helped other patients [38]. This allows patients to gather motivation for their own ED recovery [38]. To date, such ED recovery stories have not yet found a place in therapeutic ED treatment, as appropriate manuals and guidelines are still lacking [36]. However, the development of the ED recovery movement is becoming more and more important [42]. Especially on Instagram young patients come together and exchange information about their experience with ED by using hashtags like #edrecovery. Previous research on ED recovery accounts has focused less on the perspective of those affected [42].

Therefore, this study aimed to find out how ED patients use Instagram with regard to their disease and how they exchange with other patients, and if and how recovery accounts on Instagram can be an opportunity for self-help. The research question was:

If and to what extent can social media provide an opportunity for ED patients in the area of self-help?

To answer the research question, three sub-questions were developed, which include typical fields of action for community self-help: What content is communicated on Instagram ED recovery accounts compared to self-help groups? How are Instagram ED recovery accounts used by people affected and what are the effects on those affected by this usage? How does the communication and patient exchange take place on the Instagram ED recovery accounts?

The study is intended to provide an initial insight into the possibilities of using social media for self-help in this area. It also contributes to if and how online communications on social media could be helpful as a part of a self-help approach (e.g., empowering, providing hope). Self-help organizations and therapeutic personnel may use the results to examine an enhancement of existing support services.

## 2. Materials and Methods

Due to the exploratory nature of the study and the sensitive topic, a qualitative design was chosen. Semi-structured interviews with two groups were planned: people affected who are running an Instagram ED recovery account and self-help experts. To capture distinctive dimensions, increase data richness, and converge different information on the same phenomenon [43], different perspectives on the study subject were included.

### 2.1. Sampling

The sample of the interview study was divided into two groups: the self-help experts and the recovery account people. The recovery account people (in the following, also called participants) include individuals who currently are or used to be affected by an eating disorder and maintain an Instagram account with recovery content.

The population of eligible recovery account people was not known in advance. There are around one billion Instagram accounts worldwide [44]. The number of Instagram users in Germany is difficult to quantify. Searching on Instagram for hashtags such as “#edrecovery”, “#anorecovery”, “#recoveryjourney” or “#anarecovery”, a large number of posts could be found [45]. Therefore, it can be concluded that there are a similar number of accounts behind these posts. The sample of the recovery account people was determined based on the following predefined criteria: (1) An affected or formerly affected person, who is over 18 years old, owns the account; (2) the account is about eating disorder recovery content (hashtags like #edrecovery, #anorexiarecovery #anorexia recovery, etc., are used in the account description); (3) the account has at least 200 followers, so that a certain reach/influence on Instagram can be achieved and so that the content of the posted articles has a corresponding relevance; (4) it is an active account with regular activities (at least 1 post per week, or at least weekly stories); and (5) an exchange with the followers via the comment function is evident under the written posts.

A total of 19 recovery account people were contacted via direct messages on Instagram. Out of these, 11 people responded to the interview request. One person answered not being interested in the interview. To the remaining 10 individuals, the researchers sent an e-mail with further information (information and clarification letter for interview participants, consent to the processing of personal data for the research work, and a commitment to comply with data protection requirements under the German General Data Protection Regulation). An interview appointment was then arranged with six recovered account people. The duration of the interview was scheduled to be one hour.

For the sample of self-help experts, people who guide self-help groups or discussion groups for eating disorders, people from self-help contact centres, self-help information centres, and people from clinics for eating disorders were searched. In Germany, professionals, which is why no further restrictions were made for our sample, usually do not run self-help groups. The expert interviews were intended to provide additional information and corresponding contextual knowledge [46] through their observations within the self-help groups. A total of 18 individuals and institutions were contacted via e-mail. The search for experts was conducted via the Internet. The focus was limited to the federal states of Berlin, Brandenburg, and Mecklenburg-Western Pomerania due to the geographical proximity and time limitations of the study. In total, 12 people/institutions responded to the interview request. Out of these, 3 persons showed willingness for an interview and an interview was conducted with 2 persons.

The final sample (6 recovery account people and 2 self-help experts, total *n* = 8) was supposed to include typical cases with respect to the predefined criteria. The cases were selected according to their accessibility [47]. This recruitment strategy resulted in a non-probabilistic sample for both target groups.

### 2.2. Data Collection

Data collection consisted of an interview guide for the semi-structured interviews. The topics covered in the interview guide are based on the fields of action of German community self-help [48]. The emphasis is on the interviewees’ subjective perspectives [49]. In detail, the interview questions for the recovery account people consisted of four categories: (1) information on the account; (2) own use of Instagram; (3) exchange with other patients; and (4) effect on participants. An example of an open question is: “What do you observe about yourself when you use Instagram?” The interview questions to the experts consisted of three categories: (1) The role of information exchange in self-help groups; (2) patient exchange; and (3) self-help on the internet. An example of an open question to the experts is: “Please describe the role of knowledge transfer and the provision of information in self-help”. The interview guide can be found in the Appendix A.

The data were collected between June and August 2020. Due to current contact restrictions, encrypted video conversation via Jit.si was used. Interviews were tape-recorded, and subsequently verbatim transcribed [50].

### 2.3. Data Analysis

The transcripts were analysed based on qualitative content analysis [51] using the software MAXQDA (2020, VERBI Software. Consult. Sozialforschung GmbH, Berlin, Germany). Two different sets of categories were developed for the recovery account people and the self-help experts. The main categories were deductively defined according to the main topics of the interview guide, e.g., “content of the recovery accounts” and recorded in the codebook. Subcategories resulted from inductive coding based on the interview material. In the second step, cases and categories were compared with each other by means of contrasts in order to answer the research questions by means of similarities and differences. The comparison dimensions are based on the defined main categories [52].

### 2.4. Compliance with Ethical Standards

Ethical approval for the study was obtained from the Ethics Committee of the University of Neubrandenburg (reference: HSNB/GPM/156/20). The participants were informed by letter and personally about the aim of the study and the implications of participation. The participation was voluntary, and pseudonymisation was guaranteed.

## 3. Results

### 3.1. Sample Description

In total, six recovery account people and two self-help experts participated in this study (total *n* = 8). Table 1 summarises the main characteristics of the sample. On average, the interviews with the recovery account people took 58.2 min. Interviewees were on average 26 years old and had 1697 followers, all of them female. Most suffered from anorexia and bulimia. Four of the recovery account people described themselves as healed, the other two as still in recovery.

The two self-help experts were 32 and 55 years old, and the interviews took 61.5 min on average. They were both female, one of the experts was a social worker in the field of eating disorder self-help, and the other one was formerly affected by an eating disorder and now accompanies a self-help group. Table 2 summarises the main characteristics of the sample.

### 3.2. Interview Results

First, the findings from the interviews with the recovery account people are presented and then contrasted with the findings from the interviews with the self-help experts. The following is a synthesised report of the results. A total of seven main categories were identified to answer the research questions. See an overview in Table 3:

#### 3.2.1. Anamnesis, Genesis, and Usage

The recovery account people talked openly about their own medical history and the treatments they had undergone so far. Two participants were still in recovery, all others talked about their ED in the past and described themselves as healed. Almost all recovery accounts people had therapy experience and saw therapy as the cornerstone of their ED treatment. Instagram was regarded not as a substitute for conventional therapy. However, two participants felt that traditional treatment methods were not sufficient, as too little attention is still paid to the personal circumstances of the patients. Among the recovered account people, only one had experience in the field of self-help groups. This experience was evaluated negatively. Moreover, treatment support through a self-help group was rejected by this participant because a real meeting with other patients could lead to triggering situations or there is the fear that one compares oneself with other people and thus negatively influences one’s own disease progression.


*“No, I have always resisted this [self-help groups]. I couldn’t accept it for myself, because I always thought to myself: Okay, then you’re in a room with eight, nine, ten other affected people. Mostly without therapeutic guidance. And I was always not stable enough to say to myself: Okay, I would really be able to endure that”.*
(B6)

The category genesis of the account addresses the question of the recovery account people’s motivation for creating their Instagram account. For this purpose, two subcategories were distinguished: self-motivation and lack of support in own’s disease journey. Almost all recovery account people started their accounts out of their own motivation. In this context, most participants reported a lack of support in their own disease journey, as expressed by participant B5:


*“And in the past, I often wished to have an exchange with other patients, or somehow to have someone who understood me a bit”.*
(B5)

Recovery account people reported that they sought and found this support on Instagram.

#### 3.2.2. Content

The topics of the recovery Instagram accounts were very diverse, and four subcategories were identified: ED content, life and everyday life in recovery, outside of the ED topic, and taboo topics. The most important sources of information for the recovery account people’s postings were their own experiences as patients. People in recovery accounts wrote about ED content: there are posts about the causes, treatment options, eating, and consequences of ED. Another topic was life and everyday life in recovery. Almost all participants reported on their own experiences and everyday life during recovery. The perspectives of what life without an ED could bring were shown, and the challenges during recovery were also shown. In some recovery accounts, people openly shared their fears and negative feelings, such as self-hate. Furthermore, outside of the ED topic, many participants wrote about mindfulness, self-love, and generally positive thoughts as well as experiences. However, all recovery account people had taboo topics that they consciously would not address on Instagram: These include private matters, pro-ana content, weight information, or general statements of numbers and values. The self-help experts interviewed also reported this range of topics and the sharing of experiences in real self-help groups:


*“The topics are diverse, really diverse. There are hours and sessions that are more about practical things, I would say. We exchange experiences with clinics pros and cons, therapy pros and cons. I’ll say situations in the family pros and cons. Going out to eat, going swimming. The range is very, very wide”.*
(E2)

Information on calorie intake or weight data was also a taboo subject for the experts. In the sessions, the group leaders made sure that such topics were not brought up by the patients. The experts here fulfil a kind of guardian function for the group. Such a guardian function was not found on Instagram and thus poses a major difference between real self-help groups and Instagram.

#### 3.2.3. Usage

The recovery account people had two different ways of using Instagram for themselves and their ED: giving support and receiving support by sharing experiences. All participants wanted to use their account to provide support and to help other patients:


*“So, help for others I would say in any case. Just to give courage, to motivate, maybe also to be a bit of a role model, maybe to go ahead. Because yes, so it is just because I would like to be just such a person who is in the life of another, who was also affected, but who has made it. So that you just have someone in your surroundings, sort of even if only on Instagram, but who still is just with you. Who just has already managed that”.*
(B3)

In this context, the recovery account people talked about being a role model, giving courage, and showing perspectives. In addition, all participants wanted to contribute to education around eating disorders and mental diseases.

At the same time, almost all recovery account people wanted support or had received support via their Instagram account. For example, the two people who described themselves as still in the phase of recovery felt supported in relation to their eating disorder:


*“It [the personal recovery account] helps me enormously to share that. Just to talk from the soul, to get feedback somehow. But there are also a lot of people who get in touch with me. They ask about experiences and that was something I would have liked to do in the past. Or I wrote to some people. That you just know you’re not alone and other people have also gone through it. And to somehow sometimes get such an information. Because everyone knows something different, everyone has learned something different. Or already gone through this and that phase. And that is very helpful”.*
(B4)

Taken together recovery account people used their accounts to write down experiences related to their eating disorder and sharing them. They received feedback and understanding from other people via Instagram and realised through Instagram that they were not alone with their disease.

#### 3.2.4. Risks and Advantages

Using Instagram brings risks and advantages for the recovery account people. The risks include excessive time demands, content overwhelming, trigger risks, an illusory world, and limited depth of relationship.

Excessive time demands: Almost all recovery account people spoke of excessive demands in relation to the time used for their recovery account. Excessive time demands were evaluated negatively as the recovery account people sometimes could not cover all requests from their followers and, from their point of view, spent too much time on Instagram. This spreads into other areas of their lives. One person reported that she sometimes lost herself on Instagram when consuming content herself. The time that elapsed was seen as a loss and left her with an uncomfortable feeling.


*“I notice that I use Instagram more often when I’m feeling bad. Then I really lose myself in it more and more, looking for an escape for myself”.*
(B6)

Furthermore, most recovery account people reported being overwhelmed in terms of content. This overload is felt when dealing with the requests from followers. In some cases, there were situations in which the followers were perceived as needy and demanded attention from the recovery account people. They reported that followers contacted them with their psychological problems and wanted to get help. An inner conflict was reported as the recovery account people wanted to help their followers, but also knew that this was only possible to a limited extent via Instagram. The recovery account people were aware of the responsibility they had for their followers, but they tried to distance themselves from these situations or refer to professional help offers.


*“Because there were a few situations in the past where I thought to myself: This person [follower] is in so deep right now and someone has to help them. Someone has to wake them up. But I can’t do that”.*
(B4)

Trigger risks: The negative impact on Instagram for the mental health was reported as a disadvantage by almost all recovery account people. On Instagram, there would be not only recovery accounts, but also a lot of pro-ana contents as well. Other users would share content that negatively influences other patients, which can lead to relapses, according to the recovery account people.


*“And it can also be that it is a relatively good account and there are two, three things posted that trigger you, which then pull you down again”.*
(B3)

Another possible disadvantage of Instagram was the illusory world and superficiality experienced by recovery account people. On Instagram, many users show off their perfect bodies and their perfect lives. In some cases, Instagram portrays a false image of recovery, as participant B4 explained:


*“I think that’s just such a big problem that many then just hide things. And you just don’t see that when you look at the person and the Instagram account”.*
(B4)

In self-help groups, experts seek to take care to ensure that participants do not become overwhelmed or that the sessions do not have a negative impact on the patients. This protection did not exist on Instagram.

In addition, recovery account people continued to describe certain limits of the online exchange. Instagram would have limited ways to connect with each other. Partly, the signs were limited, emotions could also be conveyed only with difficulty. However, besides the risks, the usage of Instagram has also had positive effects on the recovery account people.

According to the recovery account people, the advantages of the usage of Instagram include social support and a sense of belonging, experiencing their own efficacy, disseminating information on ED recovery, accessibility, and positive anonymity. Regarding the advantages of the usage of Instagram for the recovery account people themselves it was reported to have a supportive effect in relation to their ED. This supportive effect could be seen in the social support and sense of belonging experienced by the recovery account people. They expressed the feeling of not being alone because they could exchange ideas with other people affected via their accounts. Furthermore, recovery account people reported that Instagram was a place for them to see that there were many other affected individuals who felt the same and with whom they could share experiences related to their ED. A sense of affiliation is created because they recognise themselves in other recovery accounts, people, or their experiences. Mutual motivation had great significance for the recovery account people. On Instagram, they could write about their problems and receive both understanding and feedback from like-minded people. They receive advice for their own healing process and are motivated by their followers. This is represented as follows by a quote from participant B6:


*“Because there are so many people who feel the same way as you do. And that can be totally supportive to realize that you are not alone with everything. And there is someone who has actually been through exactly the same thing as me. This person somehow survived and is now feeling better again”.*
(B6)

In addition, the recovery account people received messages that their content also helped others get better. Their helpful behaviour and their work on Instagram would be highly appreciated by their followers:


*“But of course, I also benefit from the positive feedback. And of course, it’s also really good when you realise: Okay, people like what you write. People like you. People feel inspired by you. And somehow also when someone writes to me: ‘Yes, because of you I finally dared to sign up for a clinic’, or ‘To tell my therapist about this and that’. Then I always think to myself: Yes, somehow, that does something to me. And it also has a positive effect on me”.*
(B6)

For most recovery account people, Instagram also offered the opportunity to educate on ED-related topics and eliminate prejudice and stigmatisation. For example, nutrition myths could be addressed, and background knowledge could be provided. Instagram was the perfect medium for them to reach their young target group.

There were also some advantages, which are also applicable to the other users of Instagram: time and location-independent accessibility. As access to a support network is possible regardless of place and time, that is why Instagram could be good for bridging waiting times, *“to be not left hanging in mid-air”* (B2). Another mentioned advantage was the anonymity on Instagram. For some people with ED, it would be easier to talk about emotional issues online or to ask for help online. In addition, as recovery account people explain, some followers would find it easier to contact someone on Instagram than to go to a therapist. On Instagram, one would be *“in such a protected setting”* (B6) and thus more prone to seeking help than in the existing healthcare system.

Some advantages of recovery accounts were also reported by the experts interviewed in comparison to real group sessions. The sense of not being alone and experiencing understanding would be beneficial on Instagram and in groups. The mentioned differences were the location-independent and constant accessibility of Instagram and the positive anonymity of Instagram—which locally-based self-help groups could not provide.

At the same time, experts criticised the lack of quality assurance for existing coaching offerings on Instagram. Regardless of qualification, any person could offer an online coaching or program for ED recovery via Instagram. The requirements regarding necessary certifications would have to be the same for all providers. One expert criticised that currently there would be too many people *“who make a business out of it for themselves as well”* (E1). Or as participant E1 formulates it:


*“There are just so countless coachings and programs that these formerly affected people then throw on the market. They can just do that. Without any quality management. That is also the difference to our work”.*
(E1)

It can be summarised, that self-help experts stated that similar requirements about quality assurance and corresponding structures in self-help would have to be created so that Instagram could be an opportunity. Furthermore, the formal conditions regarding data protection would have to be examined more closely, as the participation experts acknowledge.

#### 3.2.5. Self-Protection Strategies

One disadvantage of the use of Instagram was the possibility of negative influence. In the self-help groups, the experts make sure that participants set their boundaries. This protection does not yet exist on Instagram. Nevertheless, five recovery account people had developed behaviours that were intended to protect them from triggering situations or characteristics that were helpful in this regard: Only consume content that had a positive effect, unfollow content and accounts that had negative effects, took Instagram breaks, and sought support in real life:


*“Of course, you always have to look for yourself: What is good for me? Where are my limits? And I think that’s just a little bit of the hard part. Finding a healthy balance. Like saying: Okay, I have a few people that I follow and that do something good for me. But maybe also to unfollow other people who don’t do me any good”.*
(B5)

The condition for this would be a certain ability for self-reflection and one’s own motivation for therapy. It would also be important that the patient has already had his or her “click moment”. One’s own therapy motivation and the desire to really want to change something were also a condition for Instagram to be used positively, according to recovery account people.

#### 3.2.6. Communication and Patient Exchange

The patient exchange took place via private messages, Instagram posts, and contact outside of Instagram.

Private messages would be predominantly used by followers to get in contact with the recovery account people. In most cases, recovery account people were reported to be contacted by their followers in private messages and were asked for advice, opinions, or experiences. Sometimes short exchanges occurred, but none of the participants described a lively, longer-lasting exchange. The recovery account people try to address questions or topic requests from their followers via the feeds they post. The comment function was then also used in some cases for exchanges.

The patient exchange on Instagram and in self-help groups shows some similarities as well as differences. Experiences and adventures were shared both on Instagram and in self-help groups. On Instagram, mutual motivation and the feeling of not being alone played a major role, according to the recovery account people—in self-help groups, patients also benefit from this. However, the recovery account people did not use Instagram for discussions or deeper conversations, as in self-help groups. Some recovery account people reported that they were also in contact with their followers outside of Instagram. For this purpose, telephone numbers were exchanged with selected people or personal meetings took place. There was usually a more emotional relationship with these people.

## 4. Discussion

### 4.1. Self-Help vs. Recovery Accounts

First, similarities between recovery accounts on Instagram and self-help groups can be found. Our data shows that in self-help groups and on recovery accounts, the participants decide on the topics to be discussed. Sharing one’s own experience as a patient, is in both forms, the priority. As in self-help groups, the topics of the ED recovery accounts are very diverse: ED content like treatment options, life, and everyday life in recovery and topics outside of the ED are common. There are also similarities between recovery accounts and self-help groups regarding taboo topics like calorie intake, weight data, or pro-ana content. Only the lack of control over compliance with the topics is missing on Instagram and is to be evaluated negatively.

In addition, “acquiring knowledge” and “learning together” [48] in self-help groups play an important role in the accounts as well. Furthermore, comparing the use of recovery accounts with the self-help definition of the Statutory Health Insurance Funds Association [2], parallels can be seen. According to this definition, self-help is defined as a “…voluntary associations of affected people (…) whose activities are directed towards the joint management of diseases, consequences of diseases and/or also psychological problems from which they are affected either themselves or as relatives” ([2] p. 9). The field of action of community self-help includes “exchange and mutual help within the group” [48]. Exchange and mutual support take place on Instagram—not within a fixed group, but between people with the same disease. Our findings suggest that recovery Instagram shares similarities with self-help for those affected. However, to date, self-help is rarely recommended as a treatment support option in Germany. There are no evidence-based guidelines so far [17]. Through posts, recovery account people convey knowledge and gain new perspectives through other users. Thus, it can be stated that recovery accounts on Instagram fulfill the self-help idea.

### 4.2. The Usage of Recovery Accounts

Our research revealed that the recovery account people want to use their Instagram accounts to receive help and to give support to other people affected. On their recovery account, they share diverse aspects of their life during recovery, their feelings, or experiences with treatments. However, they also read posts from other users to get motivation for their own recovery. They exchange ideas, share thoughts, and emotions with like-minded people. In the preliminary considerations of this study, it was suspected that the use of recovery accounts and GSH programs share some similarities. However, GSH programs are usually facilitated by a healthcare professional [7] and are designed to allow patients to independently solve their individual problems related to their condition [8]. The interviews mainly provided information about the exchange with other patients, which is not part of the GSH programs. Moreover, GSH solutions are mostly based on cognitive behavioural therapy. Only a few parallels to GSH can be identified. The participants themselves learn new perspectives and ways of dealing with their ED and that they try to solve their problems on their own. Nevertheless, GSH programs have shown that new and more flexible treatment options can have similar effects on ED symptomatology as conventional therapies [7,9].

Significantly more similarities can be seen between recovery accounts and bibliotherapy. According to the Swedish study by Högdahl et al. [53], reading and writing texts can have a positive impact on ED symptomatology. Participants also reported a positive effect of writing and consuming ED recovery content. However, even these results can only be compared with the use of recovery accounts to a limited extent, as in the Swedish study the self-help manual was based on behavioural therapy and accompanied by a therapeutic professional [53]. The use of recovery accounts cannot be compared to conventional therapy, but some similarities to internet therapy can be identified. First and foremost, Instagram provides mobility-impaired people with the opportunity to access services [11]. Furthermore, people who cannot identify with traditional face-to-face services could benefit from them [11]. In this context, our findings suggest that recovery accounts could act as door-openers and relieve patients’ fear of conventional treatments. They offer motivation and flexible help, and it seems to be a safe place for patients. This is in line with Kendal et al.’s [54] research, which concludes that online forums can overcome barriers to seeking help. After all, participants are committed to destigmatising mental disease, educating people about nutrition myths, and exploring treatment options. ED treatment support apps have similar scopes. Like Instagram, apps are rarely based on empirically supported treatment components [55]. As with the Juarascio et al. [55] results, recovery accounts can have the potential to facilitate access to treatment or to support treatment. It is known that young women with ED symptoms try to avoid the health care system but turn to Internet-based help [56,57]. Our research revealed that the anonymity and accessibility of online support play a major role for the participants. In this context, previous research has emphasised that online support is very attractive for stigmatised individuals [33]. Even if the content of such forums is negatively oriented, they can help patients feel less lonely and increase their self-esteem [34]. Nevertheless, recovery accounts should only be an addition to professional ED treatment—therapeutic treatments cannot be replaced. Recovery accounts and self-help groups are complementary offerings.

The usage of recovery accounts also comes with some disadvantages. Just as in the findings of Spettigue and Henderson [25], participants reported negative influences and that the content on Instagram can contribute to the development and maintenance of an ED. As noted by Groesz et al. [30], the negative effects on the ED are higher among younger individuals. These negative effects are particularly high for social media platforms, whose focus is on pictures [31]—like Instagram. Vries et al. [58] reported that viewing edited photos can have negative emotional consequences for users. Our study revealed that the constant comparison with other people’s pictures on Instagram is highly critical. Even positive content can affect patients negatively. Furthermore, as mentioned by LaMarre and Rice [42], recovery accounts also provide a fixed framework for how one must recover. This representation of the recovery process can exclude those affected. However, we found certain self-protection strategies that help to protect oneself from negative influences: The first self-protection strategy is to be self-reflective and motivated for recovery. The second self-protection strategy is to only consume content that is favourable for ED recovery. Content that has a negative impact on the ED should not be consumed. The last self-protection strategy is to live in real life and not only on Instagram. People affected must seek professional support in real life. If this ability is not present, then—similar to the findings of Kleemans et al. [59]—the edited pictures of other people can have a negative impact on the satisfaction with their own body—and thus also on the ED. In the future, recovery account users should be trained in how to acquire and implement self-protection strategies on their own.

### 4.3. Importance of Patient Exchange

On Instagram, this “light version” of a patient exchange nevertheless brings some advantages for the affected people. Certainly, the online exchange was not appropriate for all patients. In the same way, real self-help meetings are not the best choice for every patient. According to the recovery account people, patients who are looking for a deeper and more emotional exchange will most likely not be able to find it on Instagram. For mutual motivation and a sense of belonging with other patients, writing private messages, or posting and commenting on posts on recovery accounts is a good alternative. There is a chance to connect with like-minded people on Instagram and get a personal insight into their ED recovery. Similar effects of social support and the sense of belonging have been noted by Eichenberg et al. [34] and Tong et al. [60]. The relationship on recovery accounts between people affected could also be compared to a peer mentoring relationship [10]. The qualitative findings of this Australian study are consistent with the findings of the present work. Our results show that participants looked for realistic examples of the recovery process. Other affected people have also gone through this. They shared their difficulties, thoughts, and information about what helped them. Moreover, it is also helpful when they see that their recovery account can also help other people affected. Our findings suggest that a closer and more trusting relationship seems to be possible between the patients. The Australian study recommends some form of monitoring or mentoring for the individuals involved [10]. Perhaps a similar “guardian” function could reduce the risks of using recovery accounts. Online components do not only provide flexibility, but also promote self-management among individuals receiving care. They feel integrated into the treatment process, which can increase motivation [61]. In some cases, recovery account people sought online help because they were missing something in the treatment process. Instagram or online programs could help fill this gap in medical care. Aardoom et al. [61] have highlighted the benefits of blended care: the online components offer great flexibility and promote self-management, while the in-person sessions enhance the therapeutic relationship. Patients gain a sense of control over their disease. The qualitative results of the positive effects of recovery stories have long been proven [36,37]. Our study clearly shows that patients experience support and motivation from the Instagram community. Several studies [39,40,41] also confirm that high motivation has a positive effect on treatment adherence, treatment duration, and on decreasing pathological eating habits. Recovery accounts can show patients a way out of an eating disorder, generate motivation for therapy, and provide other patients’ experiences with the ED.

### 4.4. Behavioural Change through Recovery Accounts?

In general, self-motivation is an important part of changing behaviour [62]. People are motivated by the success of others if they are similar to them [63]. Recovery accounts can act as a positive incentive for the people affected and lead to motivational effects.

As mentioned by Smahelova et al. [38] we also found out that the usage of Instagram accounts by the recovery account people depends on their motivation and aims for ED recovery.

Compared to Prochaska and Velicer’s [64] stages of change model, a successful treatment intervention must be adapted to the specific stage of change. Moreover, Smahelova et al. [38], have highlighted that for ED patients, the use of online services is related to the motivation and stages of their illness. Indeed, our study revealed that participants who are motivated to recover can use their Instagram account for healing.

The results of the study by Steele et al. [65] indicate that an early exploration of one’s own motivation for therapy and self-efficacy can positively influence the outcome of ED treatment. The belief in one’s own self-efficacy significantly influences the ability to change behaviour [62]. Prochaska and Velicer [64] outlined in their transtheoretical model some processes of change that help to get through the stages of change. The review by Dray and Wade [66] the transtheoretical model is also applicable in ED treatment. Our findings support some of these processes, for example, consciousness raising—the participants show the negative consequences of the ED and confront the affected person with before and after pictures. In addition, the participants are role models for their followers (environmental re-evaluation) and form a helping relationship through support [64]. These findings suggest that recovery accounts can show a first step towards health behaviour change for people with ED. According to Bandura’s learning theory and psychological modelling, the consumption of recovery account content could already lead to changed behaviour patterns. This effect is strengthened if the recipients can identify with the protagonist [67]. The participants are role models for their followers, and they can identify with them. They try to encourage and motivate them by showing how desirable life without an ED can be. Positive vision communication can be a first step toward positive future fantasies [68]. In addition, the recovery accounts represent the reality and challenges of healing. If those affected connect their future fantasies with reality and challenges, mental contrast can occur as a strategy for behaviour change [68].

In addition to these qualitative results, the quantitative effects (for example, on the symptoms) must also be examined more closely. Furthermore, the risks and disadvantages already highlighted need to be considered. Our study shows that participants can be slightly overwhelmed by time and content on Instagram. Our findings are similar to the study of Smahelova et al. [38]—when participants searched for pro-ED behaviour content, they found it on Instagram and then consumed it. Instagram can influence the ED in a positive or negative way. All recovery account people reported that Instagram had also influenced them negatively at some point, even though they wanted to consume positive content. Professional support for those affected could be a protection against negative influences.

Nevertheless, there seems to be an increased need for support for patients at this point. Recovery account people reported a lack of help during their own disease journey as a reason for starting a recovery account. The findings of the Australian study by Beveridge et al. [10] point to a potential gap in current clinical support services. Future research is needed to determine why these gaps in medical care occur and how they can be addressed. In order to incorporate recovery stories into therapeutic ED treatment, appropriate manuals and guidelines are still lacking, which represents a major research need [36]. It is important that such programs are designed to promote motivational and self-management skills and include clear instructions for behaviour change [62]. In summary, future research should attempt to investigate the different usage patterns in order to work out the best usage behaviour.

### 4.5. New Self-Help Conditions

Only two recovery account people attended a traditional self-help group during their disease journey. Self-help in Germany has experienced a decreasing usage tendency among younger people. Younger patients can no longer identify with the traditional form of self-help [18]. Self-help to reach younger patients can be an opportunity for Instagram. According to Beintner and Jacobi [3], self-help research should focus on effective programs that are designed to motivate as many patients as possible to participate. Recovery accounts can thus be described as a low-threshold service that is available regardless of time and place. According to the online survey by Mühleck et al. [69], low-threshold care services can generally positively influence the development of an ED. Similarly, Williams [4] reports that flexible and fast access to self-help materials can bring some advantages for patients. Thus, there is a chance that additional user groups may benefit who were previously unable to take advantage of time- and place-bound self-help services. Furthermore, there is no quality assurance for existing offerings on Instagram yet. The anonymity of the internet bears the risk of data fraud and misinformation [11]. The question of data protection has also not yet been clarified and will have to be discussed in future research. Recovery accounts and self-help groups, on the other hand, cannot replace professional ED treatment and should only be used as a supplement.

### 4.6. Strengths and Limitations

Our study provides the first insights into a new research field. The perspectives of people affected by ED and using an Instagram recovery account and experts working with them in real self-help groups could be taken into account to pursue our study aim.

First, the cases were selected according to their accessibility [47]. This approach limits the variation in comparability. Furthermore, only those affected who have a public Instagram account and are willing to be interviewed were interviewed. Recovery account people who have a private account or have not responded to the interview request are excluded. Likewise, this method does not include the “silent” readers on Instagram. However, it must be noted that the depth and complexity of the analysis of all the material may be lost, as the focus was mainly on the thematic aspects.

Second, the strict time limitation of the study (master’s thesis) partially restricts the approach of data collection and analysis. The data were collected between June and August 2020. The COVID-19 pandemic and the contact restrictions may have had an impact on Instagram usage. Although the participants have been using their Instagram accounts for several years, before the COVID-19 pandemic.

Third, although the sample size is small, the results of the interviews can provide information about the content, the current use, the patient exchange on recovery accounts, and the conditions for community self-help [70,71,72]. Accordingly, the sample size was appropriate for the content analysis of this target group. Due to the qualitative approach, our results are not generalisable in a statistical way, but they deliver valuable subjective insight from different perspectives of a vulnerable group.

Fourth, participants were not involved in the phase of data analysis. Therefore, misunderstandings cannot be ruled out. Inclusion would have given even more depth to the interpretation of the interviews.

However, this is the first study to provide insight from people affected by eating disorders on the use of Instagram and recovery accounts as a form of self-help for eating disorders.

## 5. Conclusions

In summary, recovery accounts, similar to internet-based, app-based, and peer mentoring programs, can provide some opportunities for ED patients. The major benefits of the recovery accounts on Instagram are the sense of belonging, flexible help, and the mutual motivation of the people affected. Recovery accounts could serve as door openers for further treatment, motivation for therapy, first step towards behaviour change, and support for existing therapies. Those looking for a deeper and more emotional exchange may not find it on Instagram. Nevertheless, all participants describe receiving support from other people affected—which is what they missed in real life. If affected people can cope with the self-protection strategies, they can use Instagram positively for themselves and their ED. Nevertheless, there is a risk of negative influence, as well as content and time overload. This study showed that recovery Instagram accounts can provide an opportunity in the area of self-help for some patients. Future research should address studies on symptomatology, the different types of use, the possibility of a guardian function, and the embedding in existing treatment programs to make the opportunities for recovery accounts as a low-threshold help usable for ED patients.

## Figures and Tables

**Table 1 ijerph-19-11334-t001:** Sample characteristics of the recovery account people (*n* = 6).

Variables	Mean (Min–Max) or *n*
Interview duration in minutes	58.2 (35–117)
Age in years	26 (19–42)
Followers on Instagram	1.697 (950–2300)
Gender	
Female	6
Medical history *	
Bulimia Binge eating Anorexia Orthorexia	3 1 5 1
Self-declaration	
Healed In recovery	4 2

* Multiple answers possible.

**Table 2 ijerph-19-11334-t002:** Sample characteristics of the self-help experts (*n* = 2).

Variables	Mean (Min–Max) or *n*
Interview duration in minutes	61.5 (50–73)
Age in years	43.5 (32–55)
Gender	
Female	2
Self-help reference	
Social worker Self-help guide (formerly affected)	1 1

**Table 3 ijerph-19-11334-t003:** Overview categories.

Genesis of the Account	Content	Usage	Risks	Advantages	Self-Protection Strategies	Patient Exchange
Self-motivation	ED content	Give support	Excessive time demands	Social support and sense of belonging		Private messages
Lack of support in own disease journey	Life and everyday life in recovery	Receive support	Content overwhelming	Experiencing their own efficacy		Instagram posts
	Outside of the ED topic		Trigger risks	Disseminating information on eating disorder recovery		Contact outside of Instagram
	Taboo topics		Illusory world	Accessibility		
			Limited depth of relationship	Positive anonymity		

## Data Availability

Not applicable.

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
