# Peer review of "“That You Just Know You’re Not Alone and Other People Have Gone through It Too.” Eating Disorder Recovery Accounts on Instagram as a Chance for Self-Help? A Qualitative Interview Study among People Affected and Self-Help Experts"

_ijerph, 2022, doi:10.3390/ijerph191811334_

Round 1

Reviewer 1 Report (Previous Reviewer 2)

Dear Authors,

as my previous evaluation of your article, I found it very good, suitable to be published in this Journal, especially now, after the minor revisions you've made.

The theme, hypothesis and especially the methodology used to study this increasing issue of ED (and other health related problems) I considered as being very suitable, respecting the scientific rigour and also adapted to the actual technological development. 

The internet and social media are part of our daily life and routine bringing advantages and disadvantages alike. Besides the way you organised and analysed the data, I appreciate the fact that finally your article offer a deeper understanding of the potential benefits and advantages that technology can bring if it correctly used.

I consider your article suitable to be published in this form, congratulations.  

Reviewer 2 Report (Previous Reviewer 1)

Thank you for considering my comments and the corrections made to the work. 

This manuscript is a resubmission of an earlier submission. The following is a list of the peer review reports and author responses from that submission.

Round 1

Reviewer 1 Report

Very interesting research. 

Page 2, 2nd paragraph. “Moreover, Beveridge et al. [10] have highlighted the effectiveness of different treat-ment support in an Australian study: a peer mentoring program (PMP). The PMP focuses on the sharing of recovery experiences between a mentor (who used to be affected by an ED) and a mentee (who is currently affected by an ED). The PMP can reduce ED symp-toms and improve the quality of life [10].” There is no need to add footnote [10] a second time at the end of the paragraph.

Page 2, 3rd paragraph

„In some cases, internet-based pro-grams can increase the motivation to change [13,14], and achieve long-term effects (13, 15], or even ensure treatment success [16].” Please correct [13,15]

Page 3, last paragraph

“4. It is an active account with regular activ-ities (at least 1 post per week, or at least weekly stories).” How long back did you check to see if posts were being published regularly?

2.2. Data collection

You can add the interview questionnaires as an attachment to the article.

Was the interview questionnaire authored or did you rely on some literature?

In my opinion, there is no need to provide information about authors' contributions in the body of the article, because on page 15 there is "Author Contributions:"

Strona 4: The semi-structured interview guides were developed by V.W. and discussed in the team (V.W. & H.J.) corresponding with the research question and the focus lies upon the subjective perspective of the interviewees [49].” oraz “The codings were selectively discussed by V.W. and H.J. and finally ana-lysed independently by V.W..”

I would move subsection 3.1 Sample description to chapter 2 Materials and Methods

Page 8, 2nd paragraph:

“Taken together recovery account people used their accounts to write down experi-ences related to their eating disorder and sharing them. They received feedback and un-derstanding from other people via Instagram and realised through Instagram that they were not alone with their disease” – No dot at the end of the sentence.

A very thorough description of the results and a developed discussion relating to the various elements of the results.

Literature - items 9, 10, 13, 14, 15. All authors are not listed.

Reviewer 2 Report

Dear Authors,

I consider your research paper original and of great value not only for the theme of ED, but also for other health affecting disorders that might be overcame through self-consciousness and motivation. I appreciate the research design and the way you approach the sample group in order to prove the advantages and disadvantages of internet. Also, the conclusions are really helpful for showing the multiple implications and influences that occur through internet and social media platforms.

I consider your paper complete and of great value for future studies, so I consider it should be published, without any changes.

Congratulations!

Author Response

Thank you for the great feedback. With our research, we hope to contribute to the health-affecting disorders. Self-consciousness and motivation can be a key factor in treatment. We want to show multiple implications and influences of social media. Best regards

Reviewer 3 Report

The publication submitted to me for review analyses issues that are currently significant and relevant. Besides the fact that there are not many studies in this area and taking into account the limitations described by the authors, it will be interesting to pursue studies in this field of study.
The content of the article is precisely and clearly structured. The aims are specific and are answered in the content. The content of sections is complete and presented in a consistent manner. The results are analysed and presented appropriately. The discussion is constructive, the conclusions are concrete.
I consider that the article is well written and suitable for submission.

Author Response

Thank you very much for the great feedback. We are also interested to see how research is developing in this still young research area. The use of social media continues to increase in the younger generation and should therefore also be considered in research. Best regards